# 3D Scanning of Surgical Specimens to Improve Communication Between Surgeon and Pathologist: A Head and Neck Pilot Study

**DOI:** 10.3390/cancers17010014

**Published:** 2024-12-24

**Authors:** Vittorio Rampinelli, Davide Mattavelli, Daniele Borsetto, Robert Kennedy, Marco Ferrari, Mattia Savardi, Alberto Deganello, Piero Nicolai, Francesco Doglietto, Cesare Piazza, Alberto Signoroni

**Affiliations:** 1Unit of Otolaryngology, DSMC, University of Brescia, 25123 Brescia, Italy; 2Department of ENT, Addenbrooke’s Hospital, Cambridge University Hospitals NHS Foundation Trust, Cambridge CB2 0SZ, UK; 3Department of Histopathology, Addenbrooke’s Hospital, Cambridge University Hospitals NHS Foundation Trust, Cambridge CB2 0QQ, UK; 4Section of Otorhinolaryngology, Department of Neurosciences, University of Padova, 35121 Padova, Italy; 5Department of Medical and Surgical Specialities, Radiological Sciences and Public Health (DSMC), University of Brescia, 25123 Brescia, Italy; 6Otolaryngology Head and Neck Surgery, IRCCS National Cancer Institute (INT), University of Milano, 20133 Milano, Italy; 7Neurosurgery Unit, Fondazione Policlinico Agostino Gemelli IRCCS, Università Cattolica del Sacro Cuore, 00168 Roma, Italy

**Keywords:** 3D optical scanners, surgical specimens, head and neck surgery, inter-specialist communication

## Abstract

This study investigated the use of 3D scanning of surgical specimens using structured light scanners (OptorLab and CronosDual) to enhance surgeon-pathologist communication.The testing on cadaver and maxillary cancer specimens aimed to optimize 3D model acquisition, with OptorLab showing superior speed and quality. Challenges included complex specimen geometry, varying surface reflectivity, and soft tissue instability. Despite these factors, the resulting 3D meshes effectively visualized surgical margins and key points. The study concluded that 3D scanning offers valuable, storable digital models with potential applications in clinical practice, education, and research, improving inter-team communication and pathological staging accuracy, though technical optimization is crucial.

## 1. Introduction

Surgery still represents a mainstay of treatment for most head and neck cancers [1]. Achieving a negative margin of resection is mostly dependent on the site of origin, histology, and surgeon experience. Along with other relevant pathological features (e.g., histology and grading), the surgical margins status concurs with the definition of the prognosis [2,3] and, therefore, the need for, and the type of, adjuvant treatment in the postoperative setting. In this view, the accuracy of the pathology report is of crucial importance, but many factors can adversely affect its reliability [4,5]. In addition to the experience of the pathologist, an inaccurate sampling process and a lack of communication between the operatory room (OR) and the laboratory are two relevant limiting factors [6,7,8]. The surgical specimen is often anatomically complex and difficult to orient, requiring multiple biopsies of surgical margins, which are all elements that hamper the work of the pathologist. Traditionally, these problems are overcome by the surgeon marking the most relevant resection margins and anatomical landmarks through ink or stitches. However, this coarse modality is potentially inaccurate [9]. A possible solution to the problem might be represented by providing intraoperative pictures, along with the surgical specimen [10], but most of the time, the surgical field is narrow, the exposure is limited, and, not infrequently, a multi-block resection is performed [11,12].

Real-time pathological feedback can be given by revising an area of positivity through frozen section histopathologic analysis. However, its technical executive aspects and tangible impact on prognosis are still debated [13]. Ideally, the presence of the pathologist in the OR would be the best option to bypass the margin assessment issues and, in particular, the communication gap between the pathologist and surgeon. However, the presence of both a surgeon and a pathologist during surgeries is often unfeasible. In the last decades, many innovations for margin assessment have been developed, including Mohs surgery, molecular analysis, non-fluorescent and fluorescent dyes, auto-fluorescent imaging, micro-endoscopy, spectroscopy, and narrow-band imaging, with promising results in specific areas of application [13,14]. However, given its complexity and variability, the margin assessment process is still an open issue, with further technical refinements still required.

Considering this background, a 3D scanning of the surgical specimen, acquired at the time of the resection, could provide an accurate, complete, and permanent model of the sample surface. On a 3D mesh model, the surgeon can, for example, digitally highlight and tag points and regions of interest, based on (possibly coregistered) radiological and intraoperative findings, allowing easy and precise pathological sampling and bypassing the inaccuracy biases of any marking modality, as well as the challenging interpretation of intraoperative pictures. Clinical interest in the direction of obtaining 3D digital models of surgical specimens emerged recently from pilot activities [15,16,17] from which the clinical significance of an improved surgeon–pathologist communication based on scan models is clearly represented [18]. However, while related works based on low-cost scanning solutions seem able to provide good quality and reliable 3D models for clinical use, they still present residual/hidden issues deserving further investigation.

Through an interdisciplinary (clinical–technological) collaboration, comprising direct acquisition activities in a challenging clinical setup, the support of high-end 3D scanners, and related expertise in the design of geometry processing algorithms for the creation and handling of quality 3D models, we take advantage of the opportunities we have to experiment and critically evaluate various technical aspects that arise in the 3D modeling of resected specimens during the surgical procedure and to reflect on the results obtained, as well as on residual problems. In particular, this study (1) provides a technical and operational comparison between a high-end industrial scanner (in a configuration similar to the one proposed in [16,17]) and a bench precision 3D scanner suited to be used in the OR, it (2) elaborates technical and clinical considerations related to complex surgical sample modeling, including both preclinical (cadaver specimens) and clinical (fresh specimens from the OR) experiments, and it (3) discusses the advantages and criticisms of our approach and suggests how to develop practical solutions to introduce in the clinical routine.

## 2. Materials and Methods

A pictorial summary of our contribution is depicted in Figure 1. Two different 3D scanning systems were considered and compared for the acquisition of 3D models of different types of specimens. The first preclinical surgical phase was accomplished using anatomical specimens obtained from cadavers. Once the scanning setup was optimized for clinical application, the two systems were used to obtain a 3D mesh of surgical specimens from patients affected by maxillary cancer. Signed patient consent forms were obtained for each case, and institutional review board approval was not indicated since virtual 3D specimen models, akin to traditional photos acquired in the OS as normal clinical practice, did not reveal any identifiable features of patients. Scanning time, technical issues, and subjective judgments were recorded during all the phases of preclinical and clinical applications.

### 2.1. Surgical Phase

#### 2.1.1. Preclinical Samples

Two fresh-frozen adult human heads were dissected in the Laboratory of Endoscopic and Microsurgical Anatomy of the University of Brescia. The arterial system was injected via the common carotid and vertebral arteries with silicone rubber (Xiameter, Midland, MI, USA) stained with red Pintasol (Mixol Red E-L3mix, Kirchheim unter Teck, Germany). Dissections were performed using high-definition and 4K endoscopes and a screen (Karl Storz^®^, Tuttlingen, Germany, and Olympus^®^, Tokyo, Japan, respectively), a high-speed drill for endonasal surgery (Anspach^®^, High Wycombe, UK), and a complete set of endoscopic and open-field surgical instruments (Karl Storz^®^, Tüttlingen, Germany). A total of 6 procedures (2 for each surgical approach) were performed, aiming to obtain the following surgical specimen:–2 × total maxillectomies;–2 × mandibulectomies;–2 × craniofacial resections.


They were subdivided into two groups (each surgical approach is represented once in each group), referred to as Group-1 and Group-2.


#### 2.1.2. Clinical Samples

Four patients affected by malignant maxillary cancers were selected as pilot clinical cases, and they were homogeneous in terms of disease, type of resection, volume, and shape of the surgical specimen. This choice was made to allow reliable comparisons between technologies and sequential improvements in the setting. Table 1 reports the notable features of the 4 cases (Table 1).

### 2.2. Acquisition Phase

Two different scanners were chosen and used for the acquisition of 3D models, both of which were considered adequate for the purpose. For the first set of surgical specimens (Group-1) and Patient 1, 3D models were obtained (1 for each cadaver and clinical procedure) with a general-purpose (industrial), high-end, structured-light 3D scanner (model: Cronos Dual 2.0 Mpx, by Open Technologies, Brescia, Italy). The scanner head comprising an LCD projector and two couples of 2Mpx cameras can be mounted on a tripod to facilitate commutation between two scanning fields. In our case, the scanning field was set to 200 mm, with a corresponding working distance of 410 mm and a point spacing of 125 μm. The target specimen was dried with blotting paper to avoid specular reflections during acquisition and placed on the plate of a single-plane, 360-degree rotating platform (Figure 2A). The captures were sent to a 3D scanner software (OpticalRevEng 4.0, Open Technologies, Brescia, Italy), which implements a multiview alignment of the single range images (RIs) acquired at each rotation step. The pre-calibration of the rotating table position allows for an expeditive coarse alignment of the RIs, followed by an optimized global registration with the method described in [19]. To fully cover the specimen surface, 2 or 3 partial models (coming from the above-described alignment procedure) were acquired, typically by placing the sample on the rotating table in different positions. Where needed, the scanner can perform an effective, fully automatic 3D feature-based alignment of freely acquired RIs according to the method described in [20]. This is particularly useful in the case of a need to complete the model with additional views (by freely moving the scanner or the sample) to fill undercuts or to fuse (coregister) partial models. However, depending on the sample tissues (presence/predominance of bone or soft tissue), deformations are likely to occur due to positioning the samples differently. This may make automatic, rather than manual, coregistration difficult, with the possible need for manual editing (e.g., the deletion of duplicate deformed portions) or, in the most problematic cases, the de facto inability to merge partial models into a single model (unless advanced deformable coregistration techniques are used). Eventually, from the aligned views, the OpticalRevEng software is used to create a quality colored mesh of the sample (whole or partial models), whose geometric quality has been further improved through a state-of-the-art mesh fixing method [21].

For the second set of surgical specimens (Group-2) and Patients 2–4, 3D models were obtained with the structured light-based Optor Lab scanner (Open Tech 3D, Brescia, Italy), a bench tool consisting of an integrated projector and platform to hold the specimen. The scanning field was set to 120 mm, with a fixed working distance. The target was placed on the integrated 360-degree rotating platform with a double axis of rotation and tilting (Figure 2B). The 3D model was acquired and edited with the Optor L software (Open Tech 3D, Italy, https://scanner3d.it/optor-l-en). This implements the same alignment and mesh creation steps described before in a way optimized by the scanner manufacturer according to user-selectable profiles, based on the desired mesh quality/weight balance. Proprietary lighting calibration and multi-view processing allow for good final mesh coloring.

Table 2 shows the technical differences between the two employed scanners, Cronos Dual and Optor Lab. In addition, for comparison purposes, the characteristics of the Einscan-SP scanner, the structured light 3D scanner used in [16,17,18,22] for purposes similar to the ones of our study, are reported as well.

### 2.3. Model Annotation

Once the scanning systems and 3D model acquisition strategies were defined, a standardized final editing process of the 3D mesh data was set up. First, the obtained 3D meshes were uploaded in .obj format to an open-source system for processing and editing 3D triangular meshes (Meshlab, the Visual Computing Lab of ISTI-CNR, Pisa, Italy). A visual inspection of the preclinical acquisitions (see the examples described in Figure 3A–D) allowed us to derive useful observations, especially concerning the suitability of acquisition devices and pipelines for representation and communication purposes. On Meshlab, surgical margins of interest and critical points were highlighted with the software’s standard tools: tags, areas, and lines. Several examples of these annotations made by surgeons are shown and described in Figure 4A–D.

## 3. Results

### 3.1. Feasibility and Technical Assessments

The capture of 3D models with the Cronos Dual 2.0 Mpx scanner in a preclinical setting (Figure 3A,B) presented some noteworthy difficulties. The geometric complexity of the anatomical piece, the non-homogeneous color, the presence of shiny areas, the instability of the soft tissues, and the absence of a multi-axial rotating platform allowing for multi-perspective acquisition, produced issues in the final mesh, such as holes (due to undercuts or the presence of reflectances) or double skins (due to small tissue motion). The average time for preclinical acquisition and editing with this technology was 15 and 35 min, respectively. The capture was significantly improved with the Optor Lab scanner, thanks to the precise setting and multiple axis of rotation and tilting of the platform, which allowed for a homogeneous and multi-angled acquisition with minimal triangular voids (Figure 3C,D). The average time of acquisition and editing was also highly reduced to 6 and 10 min, respectively.

### 3.2. Clinical Application

The cleaning of the anatomical specimens obtained from live patients was performed with gauze soaked in saline solution, and it required less than a minute for all cases. The transfer of the specimen from the OR to the adjacent room settled for the experiment was uneventful. For patient 1, the time of acquisition with the Cronos Dual scanner, editing, and mesh annotation on Meshlab was 25, 40, and 5 min, respectively. For patients 2–4, the average time of acquisition with the Optor Lab system, editing, and mesh annotation was 10, 15, and 7 min, respectively.

### 3.3. Subjective Judgments

Mesh capture and editing with 3D Cronos Dual scanner and OpticalRevEng software were defined as complex and time-consuming (A.S. and V.R.), while, with the Optor Lab scanner and software, it was defined as intuitive and rapid (A.S. and V.R.). The integrated platform of the Optor Lab scanner led to a consistent system with no need for continuous adjustment of the setting for every case. The use of Meshlab software, version 2021.10 to highlight surgical margins and critical points (Figure 4A–D) was defined as intuitive with a positive judgment from the surgeons involved (V.R., D.M., and A.D.). The pathologist (R.K.) expressed a positive opinion regarding the potential guided sampling process. In general, the colors were discernible and reliable, especially with the Optor Lab system. In the clinical setting, the technical issues of the scanning process that emerged in the preclinical phase were emphasized, highlighting the differences between the two scanning systems, in favor of the Optor Lab system. With the presence of blood further hampering the data collection, a careful cleaning of the specimen was crucial to improve the quality of the scanning. The specimens of clinical cases were presented to clinicians in multidisciplinary team meetings and helped the clinicopathological correlation process.

## 4. Discussion

This paper aims to evaluate the feasibility of 3D scanning anatomical specimens from surgical procedures as a tool to improve communication between the surgeon and the pathologist in the field of head and neck surgery. We used two different scanning systems. The first, 3D Cronos Dual with OpticalRevEng software, allowed the critical aspects of anatomical 3D scanning to be explored and identified. These included (1) 3D acquisition and editing time, (2) the need for an easy-to-set up system, and (3) the need for multi-planar acquisition, facilitated by a multi-axis rotating platform, to reduce scan voids. Through the analysis of these issues, the OptorLab hardware and software were selected, optimized, and calibrated, resulting in adequate 3D acquisition, quality, times, and modalities compatible with the OR activity. The subjective judgment of the authors, in terms of the practical usefulness of the method, was positive.

In the last 10 years, the clinical interest in 3D scanning has dramatically increased [24]. In the dental field, this technology has applications in capturing impressions for prosthetic restorations, smile design, and guided surgery [25,26]. In orthopedics, 3D scanning seems to be a natural tool to exploit for the production of customized prostheses or orthoses. However, significant problems may arise that make the usage of this technique not immediate, especially in unconstrained setups and in some pathological contexts or anatomical districts [27]. In other clinical areas, 3D scanning has been mainly applied to research settings, without reaching extensive adoption in daily practice. In orthopedic surgery, for instance, structured light scanners (SLSs) provided sufficient information to automatically identify tissue types in intraoperative scanned models of articulations using machine learning. SLSs could be used for the 3D documentation of surgical outcomes such as prosthesis component placement, screw positioning, or osteotomy alignment, as well as enabling improved surgical planning and/or guidance [28]. Pathology has seen very limited clinical use of 3D scanning. Although the routine use of 3D data in diagnostic pathology is still far from becoming a reality, potential applications related to the use of this type of data are emerging [29,30]. In fact, the pathologist orients the surgical specimen and draws conclusions on surgical margins by integrating histologic slides, macroscopic photography, surgeons’ indications, and radiographic images. In view of this, a 3D scan of the surgical specimen, acquired and annotated at the time of resection, could provide a consistent and accurate three-dimensional model of the specimen, circumventing the inaccuracy bias of other marking modalities and the difficult interpretation of intraoperative images. The high-quality 3D model can be transferred electronically in a reasonably small file size, with the potential to replace macroscopic descriptions on electronic pathology reports. Annotations could also be added to the file during the grossing process, thus integrating the key information regarding the specimen and the model together. This allows the data usually extracted from macroscopic descriptions to remain while allowing the treating surgeon or oncologist to complete interactivity [29]. On the market, there are scanning systems, different from SLS, able to acquire 3D data from surgical specimens. The Pathobin 3D System (Pathobin, Parkville, Australia), for instance, is a photogrammetry-based device [29]. The specimen is placed on a platform, and a camera captures a $360^{\circ }$ view of the object. The image capturing is fast (provided that a multi-camera setup is available), but the 3D model requires several minutes to build. Furthermore, despite the photo-realistic quality of the generated models, photogrammetry-based devices usually offer a lower geometric 3D quality than SLS devices, with lower 3D meshing resolution and quality. In addition, small objects can lead to problematic acquisition conditioning in a close-range photogrammetric setup (e.g., the impossibility of fast multi-camera acquisitions). Compared to different SLS methods (e.g., handheld systems), the scanning precision offered by our fixed SLS setup is particularly helpful when dealing with head and neck surgical specimens, typically small and geometrically complex. Indeed, handheld scanners, despite potential applications [31], cannot offer comparable scanning accuracy [27]. Furthermore, the accuracy of the system would help expand the applications of this technology beyond merely improving communication between the pathologist and the surgeon (see the final part of the discussion).

Recently, some papers have been published regarding the use of 3D SLS of the surgical specimen in the field of head and neck surgery [16,17,18,22]. Despite the same scanning principle and similar objectives to our work, there are substantial differences in workflow organization and technical aspects that need to be emphasized. In the paper by Miller et al. [22], scanning was performed by the surgeon, but the editing and annotations on the mesh were performed by the pathologist, typically the next day. In Saturno et al. [16], the specimen was sent directly to the laboratory for scanning and annotations. In both cases, the aim was to create a 3D model of the specimen that could be annotated in the context of the pathologist’s standard sampling activities. Differently, we wanted to create a system fit for the OR in terms of the space required, complexity of use, and times so that the entire process of scanning, editing, and annotation is performed by the surgeon. The aim is to highlight the areas of most interest for the surgeon to analyze, based on a process of integration between pre-surgical data (radiology and the biological behavior of cancer) and what is directly observed during an operation. We, therefore, want to provide the pathologist with a 3D map for an easier orientation of the specimen and a guided sampling process, allowing for a faster and more rational analysis, both of frozen sections and fixed tissue processing.

A typical disadvantage of SLS technology is the low-resolution texture mapping [29]. With proper calibration, however, SLS enables 3D model acquisitions with highly realistic textures and colors. In this respect, the Optor Lab scanner offers advantages over the Einscan-SP used in [16,17,18,22]. The tiltable rotating platform and a consistent scanning configuration improve the quality of acquisitions, reducing voids and artifacts due to dark spots and reflections and allowing easy color calibration. Although normal ambient light may be adequate for acquisitions, the Optor Lab can be easily shielded from light for more consistent color renditions. Finally, the Optor Lab provides accurate acquisitions for very small samples (even smaller than 1 cm^3^), while the minimum declared object size for the Einscan-SP scanner is 3 × 3 × 3 cm.

A hypothetical disadvantage of the application of 3D scanning is the increased working time for the surgeon and the laboratory in the model acquisition and editing phases. However, this could occur in the initial setup phase with users unfamiliar with the hardware and software [29]. Our results in terms of operating time and total effort (in line with [16,17,18,22]) proved to be compatible with daily activity, allowing this technology to be quickly and easily applied to routine practice in the operating room and laboratory. Moreover, the number of patients included in our and the other pilot studies [16,17,18,22] is not negligible, underlining the potential for real integration into daily clinical practice. However, to evaluate the true compatibility of this technology within the setting of everyday surgical practice, particularly for standard and short-duration cases, it will need to undergo routine testing as the subject of future studies.

The initial cost and time increase due to the introduction of new technology could potentially be followed by an overall cost reduction in the medium term, as a more precise definition of surgical margins may lead to a better selection of patients for adjuvant therapy, with the optimization of resources. Although it has not yet been made official, the cost of the Optor Lab system is expected to be in the order of 10k euros.

To sum up, Optor Lab is a scanner designed for dental applications, jewelry, industrial tooling design, or cultural heritage documentation. It was never before tested in the operating room. What emerges from this study is that it offers significant advantages: highly standardized and automated acquisition due to the internal multi-axis turntable driven by the scanning SW. This enables optimized and accurate acquisitions for each specimen and allows for the minimal manipulation of the surgical piece, a crucial aspect for use on fresh specimens in the OR. As far as a comparison of the proposed techniques against what has recently been proposed in the literature is concerned, consider the following:The complexity of our samples is higher, as was the quality of the obtained models compared to those produced in [16,17,22]. The model quality (which is also high in the case of the Cronos Dual scanner) is highly relevant for several clinical reasons: it is fundamental for a good representation of complex specimens, it allows more precise labeling and annotation (surgeon) and sampling (pathologist), and it allows for the visualization of small details that can greatly enrich surgeon–pathologist communication.The bench-top instrument Optor Lab presents several advantages compared to the high-end industrial scanner Cronos Dual and the low-cost solution proposed in [16,17,22]: more automation in the acquisition (i.e., integrated tilted axis acquisitions and stable light and color rendition), very small sample acquisition capability, and a stable and compact setup particularly desirable in an operating room context for both occupation space and acquisition reproducibility reasons (even with small optical heads and bench tripods, the setup is not as stable and reproducible compared to what a scanner like Optor Lab can offer).A slight increase in scanning times can occur due to the possibility of a multi-axis acquisition. This is to be balanced with the opportunity to better handle the specimen in order to avoid undercuts and lacking portions in the acquired surface.

Despite the potential of the technology and the favorable practical aspects of our setup, some limitations and open questions remain to stimulate future investigations. The first is related to the nature of the sample to be digitized. Soft tissue specimens may lack references that can be used to unambiguously guide the pathologist. In contrast, samples with teeth or bone components are easy to interpret. It has to be said that the same problems occur with classical sampling techniques. In any case, combining traditional orientation methods (with stitches, clips, and staining) with 3D scanning would improve the analysis process. Secondly, the possible absence of bony or cartilaginous structures supporting the sample (which depends on the sample itself) can lead to scanning errors due to position changes and the multi-axial movement of the turntable. For this reason, in our pilot study, we chose the common case of maxillary specimens, with a rigid bone structure and reduced presence of soft tissue. However, scanning errors could also be minimized in soft specimens through precise and stable positioning and the use of a rigid buttress or freezing before processing. An upcoming use of the technology will involve testing its potential on soft specimens. Thirdly, the presence of sparkles due to light reflections, excessive color contrast, the presence of blood, and complex spatial orientation may lead to holes or other topological imperfections in the final 3D mesh surface. The careful cleaning and drying of the sample is, therefore, essential to improve the quality of the scan, and the use of a state-of-the-art mesh fixing technique is highly advisable, as is the possible fusion of multiple sample acquisitions on different acquisition planes, highly facilitated by the multi-axis tilt of the OptorLab scanplate.

As a further aspect, 3D printing technology is currently widely used in medicine [32]. By illustrating an accurate and realistic representative copy of tissues, 3D-printed models of surgically removed specimens can guide surgeons during ablative and reconstructive surgical procedures [33], also assisting the training process of young surgeons [34]. To this end, a complete watertight 3D model (i.e., 3D-printable) can be obtained either (1) starting from an already almost complete model from a single (possibly multi-axis) scan, completing the model with planes or closure surfaces, or (2) from a multiple scanning process, with at least one partial model for each of the opposing surfaces of the specimen [29], followed by a fusion of the two partial models in the editing phase, based on manual or automated procedures of feature point matching, which, as we have already mentioned, could be problematic (soft tissue deformation). To this end, adequate expertise in the use of the scanner and its proprietary software is required, although these skills are quite easy to acquire, as demonstrated by the study participants.

From the perspective of integrated healthcare digitization, the potential applications of 3D scanning of oncological surgical resection specimens are manifold. They include the possibility of building a database of three-dimensional models, which could be useful for medico-legal and educational purposes [24,31,32].

Finally, 3D mesh models could be combined with preoperative imaging, allowing the correlation of margin status with the radiological aspect of a specific volume. This integration could address the challenge of the geometric relocalization of anatomopathological information [35] obtained from the specimen under surgical guidance within the operating field. Leveraging navigation systems that integrate real-time spatial information from imaging with 3D scanning data would enable targeted margin enlargemnt to be performed either during the same surgical session or in a planned surgical revision.

## 5. Conclusions

Three-dimensional SLS offers the possibility to create easy-to-store models of pathological specimens, with several potential applications in clinical practice, education, and research. The scanning process proved to be technically complex. It was necessary to undertake a process of technological optimization in order to obtain scanning modalities, times, and quality adequate for the clinical setting. The results were promising, with a potential improvement in the communication process between the surgeon and the pathologist and an enhancement of pathological staging accuracy.

## Figures and Tables

**Figure 1 cancers-17-00014-f001:**
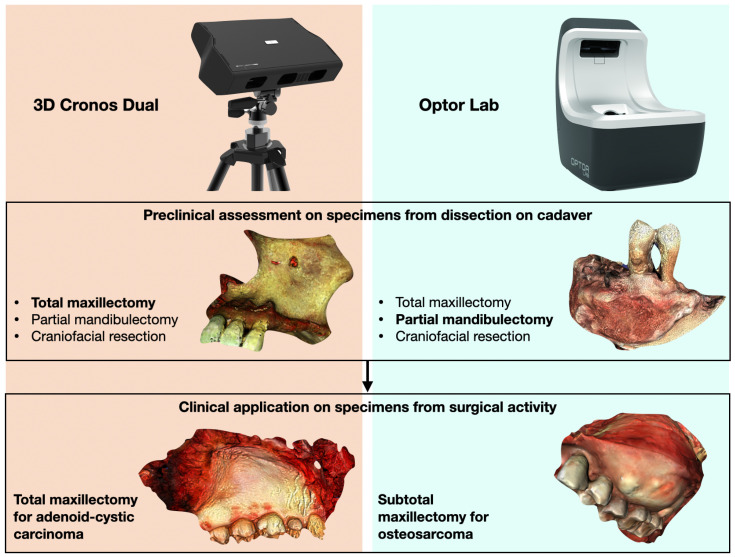
Graphical summary of the study workflow.

**Figure 2 cancers-17-00014-f002:**
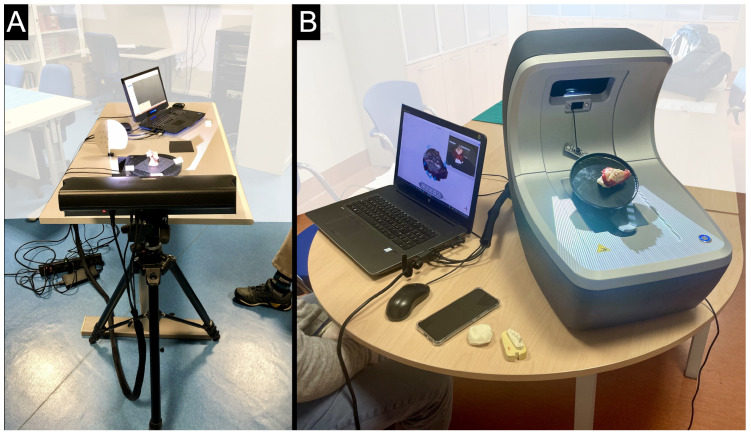
Settings. (**A**): the 3D Cronos Dual 2.0 Mpx scanner (Open Technologies 3D, Brescia, Italy) positioned at a working distance of 410 mm. The single-plane 360-degree rotating platform is visible in the center of the image. (**B**): The Optor Lab scanner (Open Tech 3D, Brescia, Italy), with the integrated 360-degree rotating platform with a double axis of rotation and tilting.

**Figure 3 cancers-17-00014-f003:**
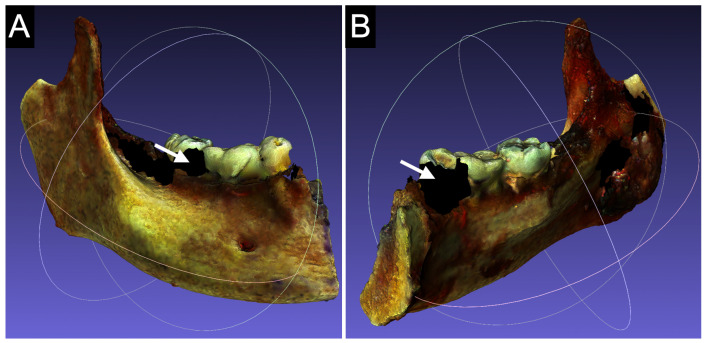
Preclinical acquisitions. (**A**,**B**): The lateral and medial face of the 3D model of a cadaver’s mandibular specimen, obtained with the 3D Cronos Dual 2.0 Mpx scanner (Open Technologies 3D, Italy). Despite the high volumetric 3D quality, the superficial texture and colors are not satisfactorily realistic. Furthermore, the single axis of rotation of the platform led to quite large areas of void acquisition (white arrows). (**C**,**D**): Anterior and posterior view of the cadaver specimen of frontal and ethmoidal bone resection, obtained with an Optor Lab scanner (Open Tech 3D, Italy). The superficial texture and colors are satisfactory, allowing for discrimination between bone, dura (white star), and ethmoidal components (the white arrow indicates the crista galli). Despite the geometrical complexity of the specimen, the multiplanar acquisition allowed by the multiaxial rotating platform led to minimal acquisition voids.

**Figure 4 cancers-17-00014-f004:**
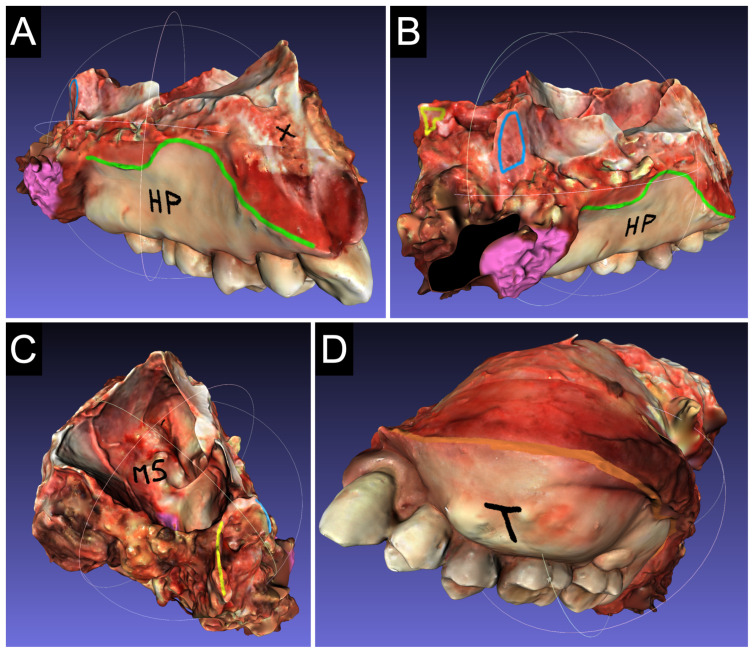
Acquisitions for ‘clinical case 2’. (**A**): Medial face of the maxillary surgical specimen from an anterior perspective. The green line highlights the medial mucosa margin on the hard palate (HP). The medial pterygoid muscle is stained pink. The black cross indicates the medullary component of the maxillary bone at its anterior margin. (**B**): Medial face with posterior perspective. The medial and lateral pterygoid plates are marked in blue and yellow, respectively. In black is shown an acquisition void. (**C**): Superior face from a posterior perspective. The tumor appears at the level of the floor of the maxillary sinus (MS). (**D**): Medial face from an anterior perspective. The tumor (T) causes swelling in the superior oral vestibule. The orange line indicates the lateral mucosal margin.

**Table 1 cancers-17-00014-t001:** Details of the 4 cases of maxillary sinus cancer included in the study. pT indicates the cancer stage of the primary lesion, obtained after the final histologic examination.

Patient Number	Histology	Surgical Intervention on T	pT	Margin Status	Scanning Tool
1	Adenocarcinoma not otherwise specified	Left subtotal maxillectomy	pT4a	Negative	Cronos Dual
2	Osteosarcoma	Left subtotal maxillectomy	pT4a	Negative	Optor Lab
3	Polymorphous adenocarcinoma	Right subtotal maxillectomy	pT3	Negative	Optor Lab
4	Adenoid cystic carcinoma	Total maxillectomy	pT4a	Negative	Optor Lab

**Table 2 cancers-17-00014-t002:** Three-dimensional scanner technical comparison: Einscan-SP (Shining 3D, China); Cronos Dual (OpenTechnologies, Brescia, Italy); Optor Lab (OpenTech3D, Brescia, Italy); and Einscan-SP scanners. ^(∗)^, according to ISO 12836 [23].

Scanner Comparison	EinScan-SP [16,17]	Cronos Dual	Optor Lab
Cameras	2×1.3 Mpx	4×2 Mpx	2×5 Mpx
Struct.light projector	white LED	white LED	RGB LED
Color acquisition	camera	camera	projector
Turntable	external	external	integrated
Tilted Angle Acq.	NO	NO	YES
Weight	4.2 kg	7 kg	22 kg
Dimensions	570×210×210 mm	540×250×145 mm	545×350×455 mm
Single scan FOV	200×150 mm	340×255 mm	150×120
Min. sample dim.	30×30×30 mm	n.d.	5×5×5 mm
Single scan resol.	∼50 μm	∼30 μm	4 μm ^(∗)^
Repeatability	n.d.	n.d.	2 μm ^(∗)^
Point distance	200 μm	120 μm	25 μm
Autocalibration	NO	NO	YES
Output formats	obj, stl, asc, ply	obj, stl, ply, off	obj, stl, ply, off
Price (approx. €)	3k	20k	10k

## Data Availability

The data that support the findings of this study are available from the corresponding author upon reasonable request.

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
