# Peer review of "3D Scanning of Surgical Specimens to Improve Communication Between Surgeon and Pathologist: A Head and Neck Pilot Study"

_cancers, 2024, doi:10.3390/cancers17010014_

Round 1

Reviewer 1 Report

Comments and Suggestions for Authors

Review of ‘3D Scanning of the surgical specimen to improve the communication between surgeon and pathologist: a pilot study in the head and neck’

By Rampinelli et al.

This paper addresses an important problem in practical surgical workflows, namely, the challenge of guiding a surgeon with respect to histopathological examination.  Overall, I found the experience of the authors with these two devices within the context of preclinical and clinical experiences interesting and worthwhile.  There are definitely some aspects that need to be added for a more complete discussion but the work is certainly interesting.

The work focuses on the creation of 3D textured topological 3D scanning of en bloc tissue specimens and subsequent labeling for assistance in communication with pathologists.  The general framework is to create a new and novel form of specimen record.  While the specific use of 3D scanning has been performed previously, the group does contribute with the comparison of two state-of-the-art scanners (with a 3rd reference based on previous literature reports).  The work also compares the results in the context of a consistent anatomical specimen and among preclinical cadavers and clinical patients.

With respect to methodology, except for the use of the Optor Lab scanning device, the work does not report any novel technology developments.  The work’s novelty comes when comparing to the more ad hoc set ups using devices like the 3D Cronos Dual, or the Einscan-SP.  The work successfully argues the point that increased integration within the workflow with better designs leads to more quality scanning.  This was illustrated in the Figure 3 result.

There are several areas in the manuscript that are lacking which if modified would lead to a much more well-rounded paper.  Detailed mapping of the en bloc specimen is only half the problem.  Creation of these digital records would allow the surgeon and the pathologist to communicate quite clearly with respect to histopathological assessment.  However, the other half of this problem is associated with the surgical defect and the need to localize within the patient any deleterious findings regarding the resection whether it be during surgery as in the case of intraoperative frozen sections or in a second surgery as in the case of a compromised margin detected post operatively.  Understanding the surgical field defect in light of pathology information is not addressed nor the challenge of that localization.  Even when surgeon and pathologist are communicating effectively about an en bloc tissue specimen, how is that information brought back to patient?  This discussion is missing.  There are not many papers on the accuracy of that process but one such relevant paper is:

Kerawala, C.J. and T.K. Ong, Relocating the site of frozen sections - Is there room for improvement? Head and Neck-Journal for the Sciences and Specialties of the Head and Neck, 2001. 23(3): p. 230-232.

The authors may want to add this into their paper with a discussion of localization.

Similar to the above point, it is important to discuss the relationship between the needed en bloc tissue 3D textured topographic specimen resolution relative to the accuracy and extent needed when a reresection takes place or when communicating pathology findings.  The authors make it a point to indicate the lesser resolutions of the more ad hoc scanner setups, but is this a problem?  Perhaps these lesser resolved scanners are already more than enough for the problem at hand?  The authors never really tie it together.  It is possible to be over resolved with a technology when practical use dictates a far lesser need.

The work does have a nice structure of preclinical cadaver followed by clinical work.  This makes for a strong comparator for the work.

The authors do indicate some aspects of deformation.  The work might be stronger if that discussion is increased.  For example, the work becomes significantly more challenging with specimens reflecting breast cancer resections.  Some aspects of the work may be more challenging than others when translating this to different soft-tissue targets.  This would be a nice discussion too.

The authors make a good argument for the more systematic design of the Optor Lab scanner.  While the Optor Lab design clearly improved timing, it would have been nice to see more discussion of the impact of timing and how realistic is this within the context of single-pass surgery.  

More Specific Critiques:

1.       Table 1 formatting is not pleasing – letters running through blocks on my print-out.

2.       Figure 1.  I do not like the descriptors on the left hand side.  You could very easily run a long block above that runs through the two right columns.  For the second caption on the left, you could do the same thing, rung a long block through the two columns on the right.  If you do this, the two white blocks with words in them would be eliminated on the left, and you could blow up the figure so I could see the devices and the quality of the scans.  Overall, the figure is too small when printed out.

3.       The white arrows in Figure 3 are subtle.  I missed them at first.

4.       Lines 199-202 are awkward and unclear.

5.       Lines 219-220 are awkward and unclear.

Reviewer 2 Report

Comments and Suggestions for Authors The paper is well structured and deals with a very interesting and innovative topic, with a satisfactory explanation of the materials and methods and repeatability. Perhaps there are few samples, but the line of research can be interesting and actionable.  

Reviewer 3 Report

Comments and Suggestions for Authors

The authors explores the idea of using 3D scanning to improve communication with pathologist in regards to communicating the specimen orientation and so on. The whole study and presentation was done well. I am in the opinion that probably the authors should be more critical of this methods by discussing (in the discussion section) the obstacles especially in regards to clinical application in more depth. Also would suggest add a discussion on the difference of using this methods compared to using an hand-held intraoral scanner used commonly in clinical practice nowadays as readers would be wondering why not use that scanner which is easier, faster, and without the need to bring specimen out of the Operating theater. Those scanners also allows pre-resection scanning which helps further in orientation of the specimen in relation to the surrounding tissues/structures.

Round 2

Reviewer 3 Report

Comments and Suggestions for Authors

I think the authors has address all previous suggestion by reviewers. The manuscript now is clearer and provides a better overall picture of this suggested method. Well done

Author Response

We here only replied to the additional request from the Associate Editor. All reply to reviewers have been already sent at Round 1. Thank you